# Impact of grain boundaries on efficiency and stability of organic-inorganic trihalide perovskites

Zhaodong Chu[1], Mengjin Yang[2], Philip Schulz [2], Di Wu[1], Xin Ma[1], Edward Seifert[1], Liuyang Sun[1], Xiaoqin Li[1], Kai Zhu [2] & Keji Lai[1]

Organic–inorganic perovskite solar cells have attracted tremendous attention because of their remarkably high power conversion efficiencies. To further improve device performance, it is imperative to obtain fundamental understandings on the photo-response and long-term stability down to the microscopic level. Here, we report the quantitative nanoscale photo-conductivity imaging on two methylammonium lead triiodide thin films with different efficiencies by light-stimulated microwave impedance microscopy. The microwave signals are largely uniform across grains and grain boundaries, suggesting that microstructures do not lead to strong spatial variations of the intrinsic photo-response. In contrast, the measured photoconductivity and lifetime are strongly affected by bulk properties such as the sample crystallinity. As visualized by the spatial evolution of local photoconductivity, the degradation process begins with the disintegration of grains rather than nucleation and propagation from visible boundaries between grains. Our findings provide insights to improve the electro-optical properties of perovskite thin films towards large-scale commercialization.

[1] Department of Physics, University of Texas at Austin, Austin, TX 78712, USA. [2] National Renewable Energy Laboratory, Golden, CO 80401, USA. Zhaodong Chu and Mengjin Yang contributed equally to this work. Correspondence and requests for materials should be addressed to X.L. (email: elaineli@physics.utexas.edu) or to K.Z. (email: kai.zhu@nrel.gov) or to K.L. (email: kejilai@physics.utexas.edu)

The worldwide surge of research interest in organic–inorganic trihalide perovskites, e.g., methyl-ammonium lead triiodide ($CH_3NH_3PbI_3$ or MAPbI_3), has led to a phenomenal increase of the power conversion efficiency (PCE) of perovskite solar cells (PSCs) from 3.8 to 22% in the past few years[1–6]. These hybrid organic–inorganic thin films are polycrystalline in nature and compatible with low-cost solution or vapor-based processes[7–9]. Yet their performance rivals many single-crystalline semiconductor solar cells[10] owing to a number of intriguing optical and electrical properties that are ideal for energy harvesting and charge transport, such as high absorption coefficient across the visible spectrum[11], high carrier mobility[11,12], and long carrier recombination lifetime[13,14]. The unprecedented progress of PSC efficiency was often attributed to the unique defect structures in the bulk and the benign grain boundaries during the early stage of PSC development[15,16].

As the PSC efficiency continues to increase, recent efforts on increasing grain sizes and/or passivating grain boundaries (GBs) have casted doubts on the general belief in the unique defect tolerance in perovskites. Several groups including us have found that, when the grain size is increased from a few hundred nanometers to the micrometer level, the device performance is often significantly improved together with elongated charge-carrier lifetimes[17–21]. At first sight, these studies imply that GBs in polycrystalline perovskite thin films may not be as benign as early studies had suggested. A recent theoretical study pointed out that GBs may even be the major recombination sites in the standard iodide based perovskites[22], which seems to be consistent with the recent experimental efforts described above. It is worth noting that regardless the chemical/physical natures of the defects, there are three primary spatial locations of defects related to perovskite thin films, i.e., film surface, bulk of the grain, and boundary between neighboring grains. Thus, in addition to possible changes of GB properties, the various new growth controls for increasing grain sizes could also affect the surface and bulk properties of perovskite grains, e.g., enhanced crystallinity, reduced defect density at the surface and in the bulk, and reduced structural defects associated with pinhole formation. Thus, it is important to scrutinize and isolate the impact of these different microscopic factors on electro-optical properties of polycrystalline perovskite thin films. Moreover, as material stability continues to be the key challenge faced by the PSC community[8], an immediate question is whether the GB and/or the surface of perovskite films are the weakest points where the degradation would start first. To this end, understanding the degradation mechanism at the microscopic level is also imperative for fabricating robust and reliable devices that meet the stringent requirements of commercialization[23–29]. In contrast to conventional macroscopic device characterizations, it is expected that spatially resolved studies on the chemical, electrical, and optical properties of the PSC thin films will provide crucial information for advancing the basic science and developing commercial products based on these fascinating materials.

While a number of scanning probe techniques[30–38] have been used to interrogate properties of the PSC, local measurements of the intrinsic photoconductivity, rather than the extrinsic photo-current across the Schottky-like tip–sample junction, have not been reported to investigate the role of various microstructures on the films. In addition, due to the poor air stability of the organic–inorganic trihalide compound, experiments on samples with a surface encapsulation layer are desirable to obtain reliable results. A non-contact method capable of mapping out the nanoscale photoconductivity on encapsulated films is thus particularly important for the PSC research. In this work, we report the first quantitative microwave impedance imaging with light stimulation on two MAPbI_3 thin films with different PCEs capped by a polymethyl methacrylate (PMMA) protection layer. The photo-response is spatially uniform across grains and grains boundaries, whereas the difference of carrier mobility and lifetime between these two films is attributed to the difference of sample crystallinity. Both the surface topography and local photoconductivity have been monitored over an extended period of time, which sheds new light on the intricate degradation process of the PSC devices.

## Results

**Microwave photoconductivity imaging on MAPbI_3.** Our experiments are performed on a unique microwave impedance microscopy (MIM)[39,40] setup with a focused laser beam illuminating from below the sample stage, as illustrated in Fig. 1a. The bottom illumination scheme avoids light being shallowed by the body of the cantilever probe[41] and ensures an accurate calibration of the areal power intensity on the sample. Taking advantage of the near-field interaction, we are able to obtain a spatial

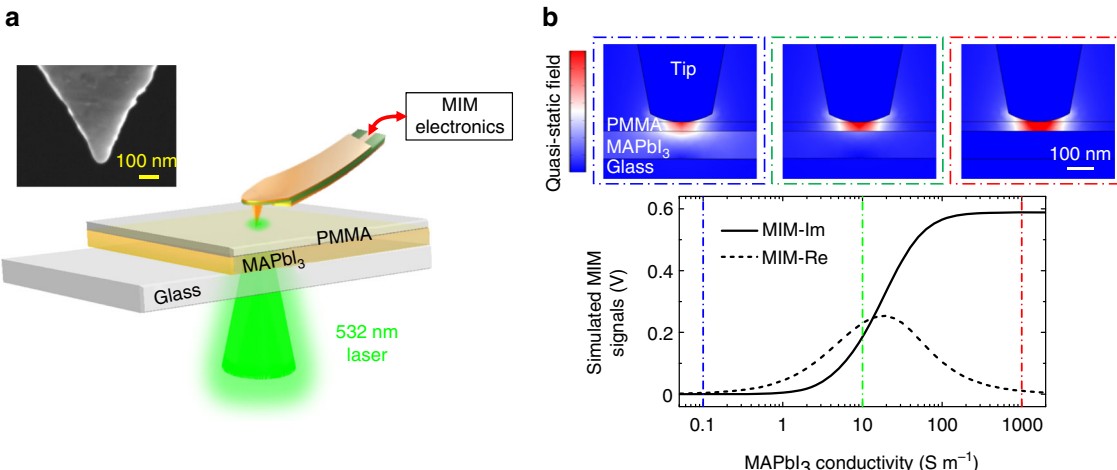

**Fig. 1** Experiment setup and finite-element analysis. **a** Schematic diagram of the sample and the MIM setup with bottom illumination through the transparent glass substrate. Scanning is accomplished by moving the sample stage while fixing the laser beam and the probe tip, which are aligned before the experiment. The inset shows the scanning electron microscopy (SEM) image of a typical MIM tip. **b** Simulated MIM signals as a function of the MAPbI_3 conductivity $\sigma$. The quasi-static displacement-field distributions when $\sigma$ is equal to 0.1, 10, and 1000 S m$^{-1}$ are displayed inside the blue, green, and red dash-dotted boxes, respectively

resolution with deeply subwavelength lateral dimensions, i.e., on the order of the tip diameter (around 100 nm or $1/30,000\lambda$, where $\lambda$ is the wavelength of the 1 GHz microwave)[40]. Here the 1 GHz signal is delivered to the shielded cantilever tip[42] and the reflected microwave is amplified and demodulated to form MIM-Im and MIM-Re signals, which are proportional to the imaginary and real parts of the tip–sample admittance[39]. Unlike the conductive AFM[32–35], a direct contact between the MIM tip and the perovskite thin films is not required due to the efficient capacitive coupling at GHz frequencies, overcoming the challenges of interrogating nanoscale electrical properties on samples with an insulating capping layer. The contrast mechanism of the MIM is similar to that of the time-resolved microwave conductivity (TRMC) experiment[11,12,43–46], a non-contact technique widely exploited to study the dynamics of photo-generated carriers, except that the time resolution in TRMC under pulsed excitation is traded for spatial resolution in MIM under continuous illumination.

The 100-nm thick MAPbI$_3$ thin films are prepared by an excess organic salt based solvent–solvent extraction method[21]. The precursor solution (30 wt%) is spin-coated on glass substrates at 6000 rpm for 25 s to form a wet precursor film, which is then transferred into a stirred ether bath for 1 min, followed by thermal annealing at 150 °C for 15 min with a petri dish covered. Solar cell devices made from the same material but with thicker film (350 nm) have demonstrated a PCE of 18% (reverse scan) under the standard Air Mass (AM) 1.5 illumination (Supplementary Fig. 1). As a comparison, 100-nm thick MAPbI$_3$ films with smaller grain size was deposited using the same process except with stoichiometry precursor and annealing at 100 °C for 10 min. Devices with small grain size and thicker (350 nm) perovskite film possess a PCE of 15% (reverse scan) (Supplementary Fig. 1). To further verify the cell performance, we also measure the stabilized power output (SPO) near the maximum power point for these two perovskite cells (Supplementary Fig. 2). The SPOs are about 17.69% and 13.1% for cells based on large and small grains, respectively. In addition, energy-dispersive X-ray spectroscopy (EDS) measurement (Supplementary Fig. 3) also suggests that there is no clear correlation between the Pb and I distribution and the grain morphology (e.g., grain vs. grain boundary) in both samples. Pb and I stay roughly constant across multiple grains within experimental noise and the ratio of I/Pb is essentially the same for both samples. Consistent with previous

studies[23–25], uncapped MAPbI$_3$ thin films degrade within several hours under the ambient light and humidity conditions, as shown in Supplementary Fig. 4. To prevent such rapid degradation, we have capped the sample surface with a thin (30 nm thick) layer of spin-coated PMMA[47] in the following experiments. As shown in the schematic in Fig. 1a, part of the PMMA/MAPbI$_3$ films is scratched away to expose the glass substrate as the reference region for the MIM imaging.

Before discussing our main results, it is instructive to review the contrast mechanism and response function of the MIM technique. Figure 1b shows the finite-element analysis (FEA) results based on the actual tip/sample geometry and material properties. Details of the FEA process are described in Supplementary Fig. 5. As the conductivity $\sigma$ of MAPbI$_3$ increases, the microwave electrical fields are gradually pushed away from this layer due to the screening effect from free carriers. As a result, the MIM-Im signal, which is proportional to the tip–sample capacitance, starts to rise as $\sigma > 10^{-1}$ S m$^{-1}$, increases monotonically as a function of $\sigma$, and saturates for $\sigma > 10^3$ S m$^{-1}$. On the other hand, the MIM-Re signal (proportional to the electrical loss at 1 GHz), peaks around $\sigma = 10^1$ S m$^{-1}$ and decreases for both higher and lower conductivities. Note that while the simulation is performed for a uniform MAPbI$_3$ layer, the span of the quasi-static electric field, which determines the lateral resolution and vertical probing depth, is set by the tip diameter. The MIM thus maps out the local variation of conductivity in the mesoscopic length scale of around 100 nm.

Figure 2a, b displays the AFM and MIM images of the samples with 18% and 15% PCEs, respectively. The large difference in grain sizes between these two films is evident from the topographic data. Without the illumination, there is little MIM contrast between the highly resistive perovskite film and the glass substrate (Supplementary Fig. 6). When illuminated by an above-gap continuous-wave 532 nm laser with a power intensity $P$ of 100 mW cm$^{-2}$ (on the order of 1 Sun at AM 1.5), the 18% PCE sample exhibits clear photo-induced MIM signals, while the effect is much weaker on the 15% PCE sample. As a control experiment, we have also performed the MIM imaging when the 18% PCE sample is illuminated by a below-gap 980-nm laser with the same power density of 100 mW cm$^{-2}$, as shown in Supplementary Fig. 6. The absence of photo-induced MIM signals with below-gap illumination further confirms the photoconductivity in the MAPbI$_3$ films.

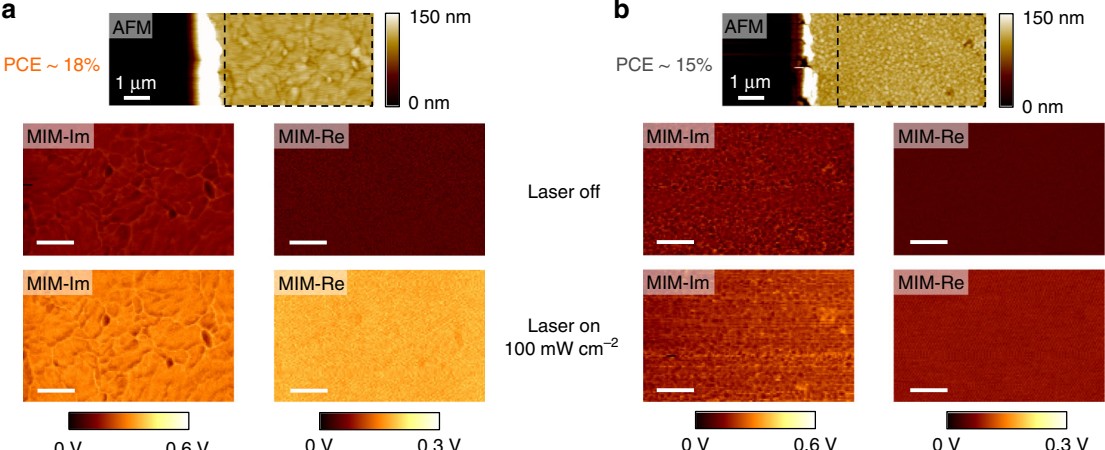

**Fig. 2** Nanoscale photoconductivity imaging on MAPbI$_3$. **a** AFM (top) and MIM-Im/Re images (inside the black dashed rectangles in the AFM data) when the 532-nm laser is turned off (middle) and on (bottom) for the 18% PCE sample. The same results for the 15% PCE sample are shown in **b**. The scale bars in **a** and **b** are 1 μm

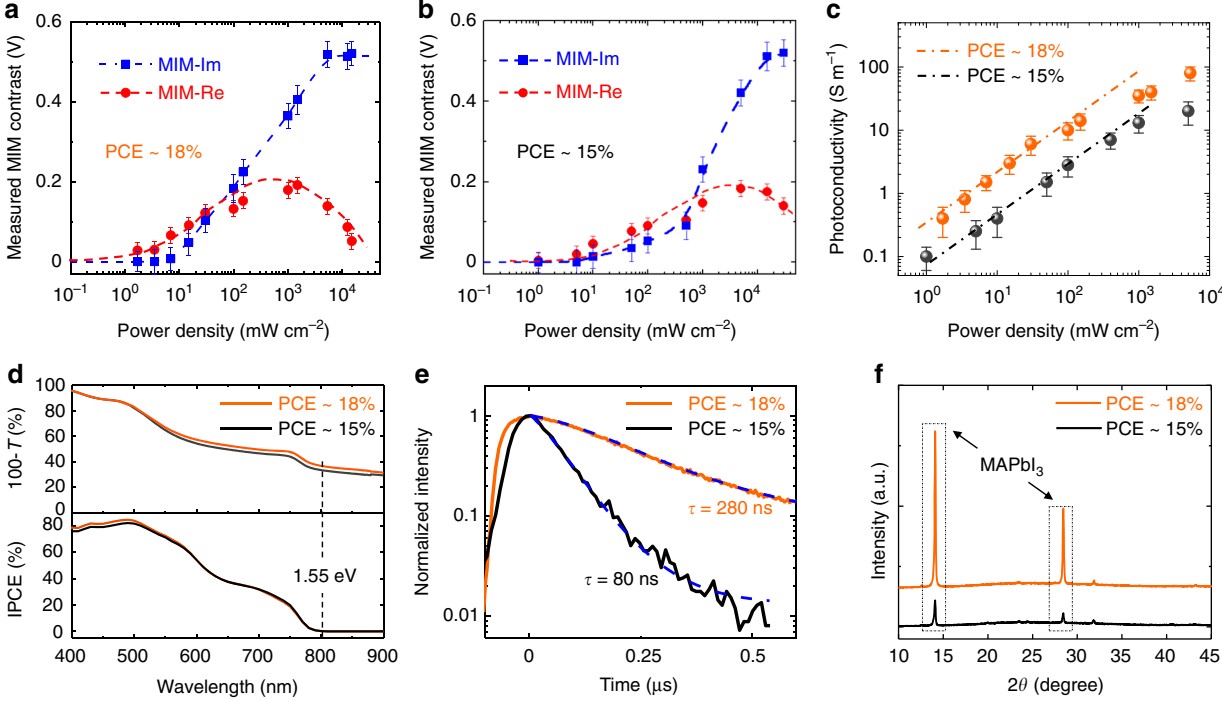

**Fig. 3** Photoconductivity and other characterizations of MAPbI$_3$ thin films. **a** Measured MIM signals of the PCE around 18% and **b** 15% samples (with respect to the values without light) as a function of the incident 532 nm-laser power. The raw data of **a** and **b** are shown in Supplementary Fig. 5. The error bars are estimated from the standard deviation of the raw MIM data. **c** Photoconductivity of both samples extracted from the experimental data vs. the laser intensity. The dash-dotted lines are linear fits for laser power below 100 mW cm$^{-2}$. **d** Optical absorptivity (upper panel) and IPCE (low panel) spectra of the perovskite thin films with 18% and 15% PCE. The optical gap of MAPbI$_3$ is indicated in the plots. **e** TRPL measurements of the two samples, from which the carrier lifetime can be extracted. The excitation power in the TRPL measurements is around 30 mW cm$^{-2}$, which is within the linear regime of the MIM data in **c**. **f** XRD measurements of the two samples, indicating a much higher crystallinity of the 18% PCE sample

**Quantitative analysis of photo-response**. The intrinsic photoconductivity images in Fig. 2 contain much information on the photon-to-electron conversion process, which is at the heart of solar cell devices. For the quantitative analysis, we first focus on the power dependence of MIM response on the 18% PCE sample. Figure 3a shows the average MIM signals over an area of 3.5 × 3.5 μm as a function of the 532 nm-laser intensity (raw data displayed in Supplementary Fig. 7a). The MIM-Im signals increase monotonically with the laser power and saturate for both $P < 10$ mW cm$^{-2}$ and $P > 10^4$ mW cm$^{-2}$. The MIM-Re signals, on the other hand, reach a peak at $P$ around $10^3$ mW cm$^{-2}$ and decrease on both sides. The close resemblance between Figs. 1b and 3a allows us to quantitatively extract $\sigma$ from the measured data. Since the film thickness is comparable to the tip diameter, the MIM is measuring the effective photoconductivity $\sigma$ averaged over the vertical direction, which can be derived as follows[48] (assuming complete absorption by the film):

$$\sigma = \frac{\eta}{H}\left(\frac{P\tau_e}{h\nu}\cdot e\cdot\mu_e + \frac{P\tau_h}{h\nu}\cdot e\cdot\mu_h\right). \qquad (1)$$

As an order-of-magnitude estimate, i.e., taking the incident photon-to-electron conversion efficiency (IPCE, hereafter denoted as $\eta$) $\eta = 1$, film thickness $H = 100$ nm, electron and hole lifetime $\tau_e \approx \tau_h \approx 10^2$ ns, photon energy $h\nu = 2.33$ eV, electron and hole mobility $\mu_e \approx \mu_h \approx 10^2$ cm$^2$ V$^{-1}$ s$^{-1}$, we obtain $\sigma \approx 10$ S m$^{-1}$ at $P = 100$ mW cm$^{-2}$, which is in excellent agreement with the data in Fig. 3c. The corresponding electron (hole) density is $n(p) = P\tau_{e(h)}/(H\cdot h\nu) \approx 3 \times 10^{15}$ cm$^{-3}$, consistent with other investigations[49]. For laser intensity above 100 mW cm$^{-2}$, the sub-linear power dependence observed in Fig. 3c may result from the saturation of photo-generated carriers[50]. On the other hand, it is generally

believed that trap states in MAPbI$_3$ remain unfilled at around 1 Sun and become filled up to 10–100 Sun under the steady-state illumination[44]. It is thus possible that trap states also play a role in the saturation of photoconductivity in Fig. 3c.

We now turn to the comparison between the two MAPbI$_3$ thin films. Following the same procedure, the MIM signals and power-dependent photoconductivity of the 15% PCE sample are (raw data shown in Supplementary Fig. 7b) plotted in Fig. 3b, c, respectively. In the regime relevant to practical solar cell applications ($P$ no greater than 100 mW cm$^{-2}$), the photoconductivity of the 18% PCE sample is five to six times higher than that of the 15% PCE sample. The two films show very similar wavelength-dependent absorptivity and IPCE over the visible spectrum, as seen in Fig. 3d. The time-resolved photoluminescence (TRPL) data in Fig. 3e show that the total electron–hole recombination time is 280 ns and 80 ns for the 18% and 15% PCE samples, respectively, from which we estimate that the average carrier lifetime differs by a factor of three between the two films. As a result, the average mobility is also improved by a considerable amount (about two times) in the 18% film. Finally, from our X-ray diffraction (XRD) data in Fig. 3f, the substantially better crystallinity of the 18% film is vividly demonstrated by its much stronger MAPbI$_3$ peak than the 15% counterpart. In all, as the average grain size increases from a few hundred nanometers to a few micrometers due to the better growth control, both the electron–hole recombination rate and transport scattering rate are considerably reduced.

**Photoconductivity of grain boundaries**. To understand the role of GBs, we directly compare the local photoconductivity between grains and GBs in the 18% PCE sample. From the AFM image in Fig. 4a, appreciable inhomogeneity is seen on the polycrystalline

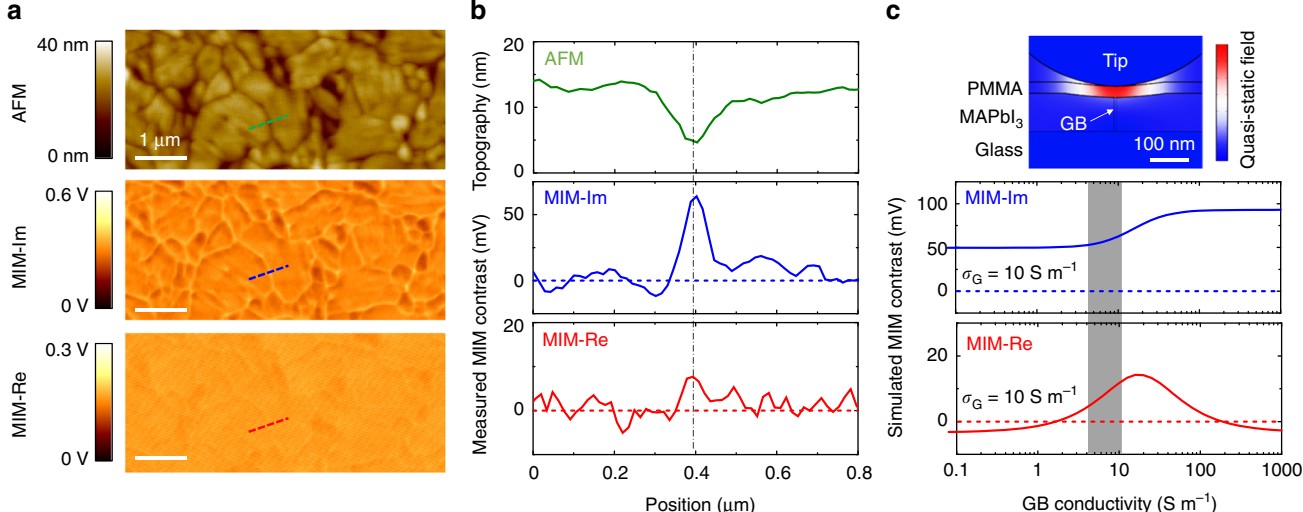

**Fig. 4** Various microstructures on the perovskite film. **a** AFM and MIM images showing the detailed features on the MAPbI$_3$ thin film under 100 mW cm$^{-2}$ illumination of the 532 nm laser. The photoresponse is uniform for grains with different heights and lateral sizes. All scale bars are 1 μm. **b** Line profiles across a single grain boundary (green in AFM, blue in MIM-Im, and red in MIM-Re) in **a**. The MIM contrast signals are referenced to the mean values on the grains. **c** Simulated MIM contrast, referenced to the background grain with a conductivity of 10 S m$^{-1}$, as a function of the GB conductivity $\sigma_{GB}$. The shaded column corresponds to the measured data within the experimental errors. The inset shows the simulated quasi-static displacement-field distribution for $\sigma_G = \sigma_{GB} = 10$ S m$^{-1}$

MAPbI$_3$ thin film, with grain sizes ranging from sub-micrometer to a few micrometers and GBs appearing as trenches of about 10 nm in depth. In contrast, the corresponding MIM images are relatively uniform over grains with different heights and lateral sizes. In Fig. 4b, we compare the line profiles across a single GB from the AFM and MIM images. Note that the MIM-Im channel is susceptible to the topographic crosstalk, while the MIM-Re signal is less sensitive to the surface roughness. We emphasize that the topographic mixing is not proportional to the AFM data and cannot be removed by simple image processing. The AFM data represent the position of the tip apex when scanning on a surface with certain roughness. The topographic mixing in the MIM-Im data, on the other hand, is mostly due to the stray capacitance from the non-apex part when the tip tracks the sample surface. For quantitative analysis of the GB photo-conductivity $\sigma_{GB}$, we perform the 3D FEA simulation (detailed in Supplementary Fig. 8), in which the GB is modeled as a thin slab (10 nm in width $w$) with a height difference $\Delta H$ of 10 nm lower than the grains (10 S m$^{-1}$). Within the experimental uncertainty, a comparison between the MIM data and the FEA shows that $\sigma_{GB}$ is within a factor of two from $\sigma_G$. As a control experiment, we have also conducted the MIM measurement on a thicker (300 nm) film of the 18% PCE sample and observed the same results (Supplementary Fig. 9), which indicates that the MIM signals are not dominated by the possible surface recombination in the 100 nm film. The result of nearly uniform photoconductivity is striking since GBs contain a large amount of dangling bonds, which usually lead to trap states detrimental to carrier generation and charge transport in conventional solar cell materials[7]. For halide perovskites, the role of GBs has been under debate over the years. While some theoretical studies showed that GBs only create shallow in-gap states due to the high ionicity and strong Pb–I anti-bonding[15,16], others indicated that GBs are the major recombination centers and should be passivated[22]. Our measurements provide direct experimental evidence that, at least for encapsulated thin films with micrometer-sized grains, GBs are relatively benign to carrier generation and transport. Perovskite films with better crystallinity, on the other hand, should be the direction to further improve the efficiency of PSCs.

**Spatial evolution of the degradation process**. The air stability of the perovskite films, which is arguably the most serious bottle-neck towards commercial applications[23–29], has also been studied by the MIM. Figure 5a shows selected AFM and MIM images of a PMMA-coated MAPbI$_3$ film over 1 week under the ambient condition with a relative humidity of 35% at 23 °C (complete set of data included in Supplementary Fig. 10). Little change in the photoconductivity is observed on the first day or two, suggesting that the PMMA capping layer was initially effective in preventing the film degradation. Starting from Day 3, however, appreciable reduction of the photo-response is seen in certain regions of the film. Interestingly, the degradation does not emerge from GBs but is associated with the disintegration of large grains, which sub-stantially reduces the photoconductivity in the surrounding regions. The degraded areas continue to grow during the sub-sequent days and the photoconductivity completely diminishes after Day 6. We note that recent transmission electron micro-scopy (TEM) studies have revealed the existence of twin boundaries that are not seen in traditional SEM or AFM ima-ges[51]. Our results are thus only valid for the GBs with clear topographic features. As a control experiment, we also con-tinuously illuminate another sample for 9 h (same as the total imaging time in the aging experiment above) and observe no change in both topographic and photoconductivity maps (Sup-plementary Fig. 11). The illumination by itself, therefore, is not sufficient to cause the drastic suppression of photoconductivity in Fig. 5a.

To gain further insight on the aging process, we have carried out structural and optical characterizations on films prepared by the same method and exposed to the same conditions. As seen in the XRD results in Fig. 5b, the characteristic PbI$_2$ peak at $2\theta = 12.66°$ appears on Day 1 and becomes significant on Day 3. The spectrum on Day 6 is dominated by the strong PbI$_2$ peak, indicative of the nearly complete decomposition of MAPbI$_3$ after 1 week. Note that the emergence of PbI$_2$ reduces the local photoconductivity because its absorption edge at 525 nm[52] is at higher energy than our 532 nm laser. The TRPL data plotted in Fig. 5c also demonstrate the same trend. While little change is observed after one day, the absolute PL intensity drops

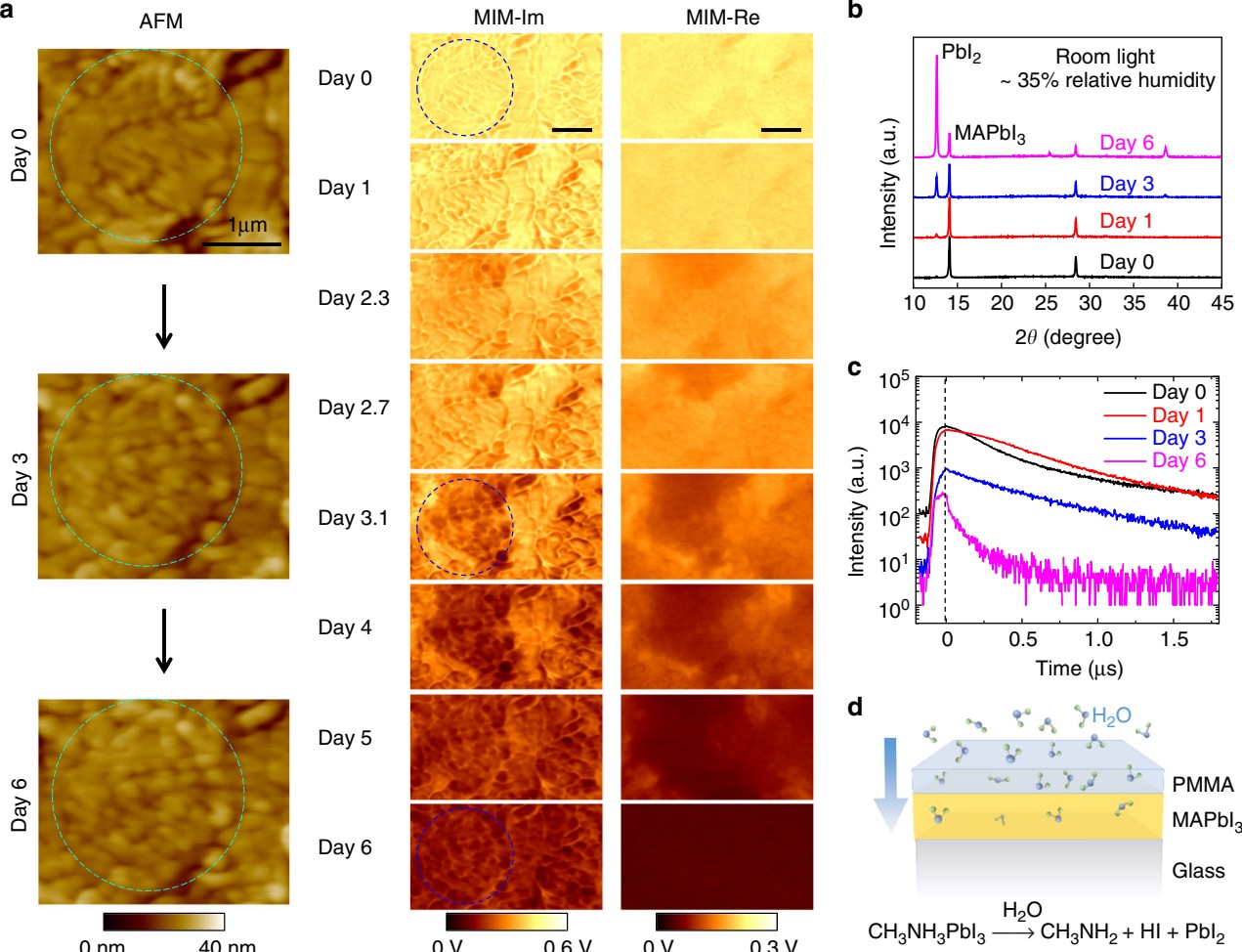

**Fig. 5** Degradation process of the MAPbI$_3$ thin film. **a** MIM images over 1 week with 35% relative humidity at 23 °C. Zoom-in AFM images on Days 0, 3, and 6 are shown to the left. Note that the fine features on large grains on Day 0 (inside dashed circles) are associated with crystal edges or terraces, which are common for polycrystalline thin films. The sample was only illuminated by the 532-nm laser during the MIM imaging for about 30 min at each frame. All scale bars are 1 μm. **b** Selected XRD and **c** TRPL data on MAPbI$_3$ films prepared by the same method and exposed to the same conditions. The XRD peak at $2\theta = 12.66°$ is associated with PbI$_2$. **d** Schematic of the aging process due to the diffusion of water molecules through the PMMA layer, which drives the decomposition of MAPbI$_3$ into CH$_3$NH$_2$ and HI in the gas phase and PbI$_2$ in the solid phase

substantially on Day 3 and the TRPL decay rate increases rapidly on Day 6. Interestingly, the TRPL decay rates are similar for the first three days, suggesting that the carrier lifetime remains large in the unaffected regions. Combining the microscopic and macroscopic results, we conclude that the deterioration of the encapsulated film is initiated and accelerated by the water molecules slowly diffused through the PMMA layer, which drive the decomposition of large MAPbI$_3$ grains into PbI$_2$ in the solid phase and CH$_3$NH$_2$ and HI in the gas phase[23–26]. Again, GBs are not the nucleation centers during the degradation.

## Discussion

To summarize, we report the intrinsic photoconductivity mapping of thin-film MAPbI$_3$ with different PCEs under above-gap illuminations. The large photoconductivity of the higher efficiency sample is a direct consequence of the high carrier mobility and long lifetime because of the improved crystallinity. Surprisingly, the GBs exhibit photo-responses comparable to the grains, and they are not the nucleation centers for the degradation process. Our results highlight the unique defect structures responsible for the remarkable performance of PSC devices, and address the significance of crystallinity to further improve their

energy conversion efficiency. We also emphasize that the nanoscale photoconductivity imaging by microwave microscopy represents a new methodology in optoelectronic research. With future development to incorporate broadband illumination, variable temperatures, and humidity-controlled environment, we expect to obtain further insights on the intriguing photo physics and air sensitivity of the hybrid perovskite materials.

## Methods

**Preparation of MAPbI$_3$ perovskite thin films**. The perovskite films in this work were deposited on top of cover glasses using the stoichiometry and non-stoichiometry solvent–solvent extraction method[21]. In short, 30 wt% methylammonium iodide (MAI) and lead iodide (PbI$_2$) (MAI/PbI$_2$ = 1/1, MAI/PbI$_2$ = 1.2/1) were dissolved in a mixed solvent of 1-N-methyl-2-pyrrolidinone (NMP) and γ-butyrolactone (GBL) (NMP/GBL = 7/3 weight ratio). The cover glass with 80 μl precursor solution was spun at 6000 rpm for 25 s, and immediately dipped into a vigorously stirred diethyl ether bath for 1 min. The perovskite films rapidly crystallized during the bathing process and were further annealed (stoichiometry precursor: 100 °C for 10 min; non-stoichiometry precursor: 150 °C for 15 min) with a petri-dish covered to remove excess organic salt. Within the experimental errors of our characterization tools, these two types of perovskite films show the same MAPbI$_3$ perovskite composition and crystal structure and only differ in grain size and crystallinity[21]. The encapsulated perovskite films were capped with PMMA ($M_w$ about 120,000) film by spin-coating 15 mg ml$^{-1}$ PMMA in chlorobenzene solution at 4000 rpm for 35 s.

**Structural and optical characterizations**. The absorption spectra were measured by a UV/Vis spectrometer (Cary-6000i). The perovskite structure was characterized by an X-ray diffractometer (Rigaku D/Max 2200) using the Cu Kα radiation. The incident photon-to-electron conversion efficiency (IPCE) experiment was carried out in a solar cell quantum efficiency measurement system (QEX10, PV Measurements). The time-resolved photoluminescence (TRPL) measurements were conducted in a home-built time correlated single photon counting system, where light source is a Fianium Supercontinuum high power broadband fiber laser (SC400-2-PP). The excitation wavelength was 500 nm with a spot area of $300 \times 300$ μm, a power of about 5 μW, and a repetition rate of 0.1 MHz.

**Microwave impedance microscopy and finite-element analysis**. The MIM in this work is based on a modified ParkAFM XE-100 platform with bottom illumination. The customized shielded cantilever probes[40] are commercially available from PrimeNano Inc.

Finite-element analysis is performed using the AC/DC module of commercial software COMSOL4.4. Supplementary Figure 5 shows the FEA simulation result using the tip/sample geometry and material properties[39]. The upper part of the tip is modeled as a truncated cone with a half-angle of 15° and a diameter of 200 nm at the bottom surface. The radius of curvature of the tip apex (modeled as a truncated sphere), 250 nm in this case, is determined by calibration on standard samples. The relative dielectric constant $\varepsilon$ of MAPbI$_3$ is assumed to be 60, consistent with that reported in the literature[7]. From the AFM data, the thicknesses of the PMMA ($\varepsilon =$ 3, http://webhotel2.tut.fi/projects/caeds/tekstit/plastics/plastics_PMMA.pdf) and MAPbI$_3$ are 30 nm and 100 nm, respectively. Supplementary Figure 5a shows the simulated real and imaginary parts of the tip–sample admittance as a function of the conductivity of the MAPbI$_3$ layer, using the values at $\sigma = 0$ as the reference points. The MIM electronics is calibrated in that an admittance contrast of 1 nS corresponds to an output voltage of 3.5 mV. In general, the MIM-Im signal, which is proportional to the tip–sample capacitance, increases monotonically as a function of $\sigma$. The MIM-Re signal, which corresponds to the electrical loss at 1 GHz, peaks at intermediate $\sigma$ and decreases for both good conductors and good insulators. The quasi-static potential and displacement field distributions around the tip–sample junction are shown in Supplementary Fig. 5b for three characteristic conductivities at 0.1, 10, and 1000 S m$^{-1}$, respectively.

The 2D axisymmetric simulation in Supplementary Fig. 5 does not work for the modeling of grain boundaries (GBs) because of the lack of rotational symmetry. To quantitatively understand the experimental data, we have performed 3D simulation and modeled the GB as a thin slab sandwiched between two adjacent grains. The center of the GB is 10 nm ($\Delta H$) lower than the surrounding grains, giving rise to the topographic artifact to the MIM signals described below. The width ($w$) of the GB, within which the electronic conductivity may differ from $\sigma_G$ due to dangling bonds and other defects, is assumed to be 10 nm. The simulated MIM signals when the tip locates on top of the GB are plotted in Supplementary Fig. 8a, together with the signals when the tip locates on top of the grains ($\sigma_G = 10$ S m$^{-1}$ under the 100 mW cm$^{-2}$ illumination). The tip–sample interaction at various $\sigma_{GB}$'s can be visualized by the quasi-static displacement field distribution in Supplementary Fig. 8b. The relative MIM contrast (GBs vs. grains) in our experiment corresponds to the difference between solid and dashed lines in Supplementary Fig. 8a. Note that the MIM-Im contrast is considerably affected by the topographic crosstalk, as the GB signals are always higher than that within the grains for all $\sigma_{GB}$'s. Nevertheless, by comparing the FEA results in Fig. 4c and the experimental data (60 mV contrast in MIM-Im and 10 mV contrast in MIM-Re) in Fig. 4b, it is evident that $\sigma_{GB}$ is not substantially smaller than $\sigma_G$ (within a factor of 2). We therefore conclude that $\sigma_{GB}$ is not substantially smaller than $\sigma_G$ (within a factor of 2). We therefore conclude that GBs are intrinsically benign in MAPbI$_3$ thin films with photoconductivity comparable to that within the grains[15].

**Data availability**. The data that support the findings of this work are available from the corresponding author upon reasonable request.

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

## Acknowledgements

The work at UT-Austin led by X.L. and K.L. is supported by NSF EFMA-1542747. Z.C. and D.W. also acknowledge the support from Welch Foundation Grant No. F-1814. The work at the National Renewable Energy Laboratory was supported by the U.S. Department of Energy under Contract No. DE-AC36-08-GO28308. K.Z., M.Y., and P.S. acknowledge the support by the hybrid perovskite solar cell program of the National Center for Photovoltaics funded by the U.S. Department of Energy, Office of Energy Efficiency and Renewable Energy, Solar Energy Technologies Office.

## Author contributions

K.L., X.L., and K.Z. conceived the project. M.Y., P.S., and K.Z. prepared samples and performed optical measurements. Z.C., X.M., and J.S. conducted the MIM measurements. Z.C. and K.L. performed data analysis and drafted the manuscript. The manuscript was written through contributions of all authors. All authors have given approval to the final version of the manuscript.

## Additional information

**Competing interests:** K.L. holds a patent on the MIM technology, which is licensed to PrimeNano Inc. for commercial instrument. The terms of this arrangement have been reviewed and approved by the University of Texas at Austin in accordance with its policy on objectivity in research. The remaining authors declare no competing financial interests.

