## [Peer Review File · Nature Communications]

Reviewers' comments:

Reviewer #1 (Remarks to the Author):

The authors have provided a comprehensive reply to the questions raised and also presented a greatly improved manuscript.

In respect to my comment comment #1-2 (3 process parameters have been changed in order to change grain size) I still feel that this is somewhat unfortunate. Grain sizes can easily be controlled simply by changing one process parameter (i.e. sintering time or temperature). At the same time I agree with the authors that the XRD does not show any major compositional differences. As such I believe that it is appropriate to compare these samples.

I would propose that the authors are more honest about the efficiencies of their devices. The '18% solar cell' appears to have less than 14% when scanned in the opposite direction. The actual (stabilised) power output must be somewhere in between these 2 values. Please state the scan direction dependent efficiencies in the main manuscript.

Apart from this minor revision proposition I believe that the revised manuscript represents some excellent work that should be published in Nature Communications.

Reviewer #2 (Remarks to the Author):

I believe the revised manuscript resubmitted to Nature Communications is now vastly improved over their previous version, especially the paragraph added covering an introduction to the technique. I can recommend publication provided the following points are addressed:

1. Some more of the discussion given in the authors' reply about the cross-talk and the limitations it imposes on extracting further information should be included in the main text. At present there is a brief sentence on it but I think this should warrant a few more sentences. Two referees had questions about this so it is very likely that other readers will too.
2. The comment that the two films have precisely the same composition and that only the grain size and crystallinity are changing is not necessarily correct. This is a problem that perovskites face more generally — in that the grains self-assemble based on the precursors and recipe conditions but do not adhere to their starting stoichiometry. It is difficult to predict the final compositions arising from each recipe due to the complicated simultaneous assembly, crystallization and evaporation of solvent, excess precursors etc during spinning and annealing. This would need to be done by direct measurement of the final films. The final compositions, particularly locally, may vary somewhat from the desired final compositions or even the initial stoichiometries. For example, one recipe used excess organic and there is no reason that some amount of this excess doesn't remain in the film. Therefore, the statement 'These two types of perovskite films have the same MAPbI₃ perovskite composition and 27 crystal structure and only differ in grain size and crystallinity²³' needs to be revised in light of this.
3. I am not entirely satisfied with the response to my original point #2-3 on the impact of traps. I agree that techniques probing mobile carriers and photoluminescence in principle can be probing different population sub-sets. However, what is true is that mobile carriers and radiative recombination will both be impacted by the presence of unfilled trap states (in different ways but still generally negatively impacted). It is generally agreed that these trap states are unfilled at around 1 sun at CW steady-state illumination and even up to 10-100 sun (where they become filled and effects saturate), so over the range of excitation densities used here traps will play a large role. Recent time-

resolved microwave conductivity measurements give further evidence to the impact of traps on conductivity (eg. doi: 10.1021/acs.jpcclett.5b01361). Is it simply a coincidence that the excitation-dependence microwave response in this work follows a very similar trend to the excitation-dependent photoluminescence trends? I think there must be a connection between them (i.e. the impact of traps) and this should at least be discussed and considered in the authors' interpretations.

Reviewer #3 (Remarks to the Author):

Thanks for the authors to address them. I think the authors basically argue on every comments I have and did not actually address them. This topic covers a very debated question. I have strong concern the conclusion in this manuscript can be very wrong without careful study. I donot see a strong evidence from this study that can make this very strong conclusion grain boundaries in perovskites are benign. The authors relied on a single measurement without considering much about material science. So I donot suggest the publication of the very preliminary study. The detail comments are given below.

1. The authors never consider the effect of film surface. It is well established in this field that the surface of perovskite films are very defective. It is true that a high efficiency over 18% can be made with these defective surfaces, because most of charge transport layers have passivation effect. The film used for this study is only 100 nm. The authors can have a simple estimation how fast the carriers (or maybe hot carriers) can diffuse to the surface and then be quenched based on the mobility they measured. There is no much difference of carrier mobility in on single grain and single crystals. If the surface recombination cannot be excluded, the authors cannot see any difference in the grains and grain boundaries, because surface recombination can be faster than that at the grain boundaries.
2. If the author wants to make a claim on discovery of correlation between crystallinity and electrical properties, the authors should have strong evidence on the crystallnity change of grains. Now every claim is based on assumptions. Regular Regular XRD generally gives grain size information, but cannot tell defect density in bulk grains.
3. I donot see how the authors can reconcile the conflicting of the statement that defects in bulk affect material property but not at grain boundary. From fundamental material science, the defects in grain boundary is a collection of point defect in the bulk.

Dear Editor,

Thank you very much for sending us the review reports of our manuscript (NCOMMS-17-08434-T) titled “Impact of Grain Boundaries on Efficiency and Stability of Organic-Inorganic Trihalide Perovskites”. Following the reports, we have performed additional measurements and revised the manuscript (highlighted in the text) accordingly. Reviewers’ comments and questions are addressed in detail below.

Reply to Referee 1’s report:

We are pleased that the Reviewer appreciates our effort to “provide a comprehensive reply to the questions and also present a greatly improved manuscript”. We are also grateful to his/her explicit recommendation of publication – “I believe that the revised manuscript represents some excellent work that should be published in Nature Communications”. The minor concern raised in the report has been addressed as follows.

Referee comment #1: In respect to my comment comment #1-2 (3 process parameters have been changed in order to change grain size) I still feel that this is somewhat unfortunate. Grain sizes can easily be controlled simply by changing one process parameter (i.e. sintering time or temperature). At the same time, I agree with the authors that the XRD does not show any major compositional differences. As such I believe that it is appropriate to compare these samples. I would propose that the authors are more honest about the efficiencies of their devices. The ‘18% solar cell’ appears to have less than 14% when scanned in the opposite direction. The actual (stabilized) power output must be somewhere in between these 2 values. Please state the scan direction dependent efficiencies in the main manuscript.

Reply #1: We agree with the reviewer that the solar cells in this study have significant hysteresis. The IV curves with both the reverse and forward scan directions along with the PV parameters are provided in Figure S1. The device with larger grain sizes showed a PCE of 18.06% under reverse scan and 13.79% under forward scan. In contrast, the cell with smaller grain sizes showed PCEs of 15.18% and 10.51% for the reverse and forward scans, respectively. The hysteresis is common in perovskite solar cells, especially when compact TiO₂ is used as the electron transport layer. To further verify the cell performance, we have also measured the stabilized power output (SPO) near the maximum power point. As requested by the reviewer, we have now provided this result in Figure S2. The SPOs are about 17.69% and 13.1% for cells based on large and small grains, respectively. These values are indeed between the reverse and forward scans results but are closer to the reverse scan results.

Changes made: We still refer to the two samples as “18% PCE” and “15% PCE” films after listing the efficiencies in the main text and Figures S1 and S2. On Page 6 of the revised manuscript, we follow the Reviewer’s instruction and state the scan direction in which the efficiencies are measured. The SPO measurements are also described on Page 6 as “To further

verify the cell performance... The SPOs are about 17.69% and 13.1% for cells based on large and small grains, respectively.”

Reply to Referee 2’s report:

We thank the Reviewer for reporting that our manuscript “is now vastly improved over their previous version, especially the paragraph added covering an introduction to the technique.” The additional points raised by him/her are thoroughly addressed as follows.

Referee comment #2-1: Some more of the discussion given in the authors’ reply about the cross-talk and the limitations it imposes on extracting further information should be included in the main text. At present, there is a brief sentence on it but I think this should warrant a few more sentences. Two referees had questions about this so it is very likely that other readers will too.

Reply #2-1 and Changes Made: We totally agree with the Reviewer that some discussions in our previous response letter about the topographic cross-talk should be included in the main text. Three sentences are now added on Page 10 of the revised manuscript “We emphasize that the topographic mixing is not proportional to the AFM data... The AFM data represent the position of the tip apex ... The topographic mixing in the MIM-Im data, on the other hand, is mostly due to the stray capacitance...” The text then naturally transitions to the description of our 3D FEA simulation. We believe that the readers can now appreciate the limitations and the need to perform careful analysis on the data.

Referee comment #2-2: The comment that the two films have precisely the same composition and that only the grain size and crystallinity are changing is not necessarily correct. This is a problem that perovskites face more generally — in that the grains self-assemble based on the precursors and recipe conditions but do not adhere to their starting stoichiometry. It is difficult to predict the final compositions arising from each recipe due to the complicated simultaneous assembly, crystallization and evaporation of solvent, excess precursors etc during spinning and annealing. This would need to be done by direct measurement of the final films. The final compositions, particularly locally, may vary somewhat from the desired final compositions or even the initial stoichiometries. For example, one recipe used excess organic and there is no reason that some amount of this excess doesn’t remain in the film. Therefore, the statement ‘These two types of perovskite films have the same MAPbI₃ perovskite composition and crystal structure and only differ in grain size and crystallinity’ needs to be revised in light of this.

Reply #2-2: We agree with the reviewer that the final perovskite composition could be different depending on the exact processing conditions. It is known that perovskite MAPbI₃ is not thermally stable. The organic component MAI can be released with excess PbI₂ leaving behind in the film. This has created a challenge to vary grain size by simply adjusting annealing temperature and annealing duration as mentioned by Reviewer 1. In a previous paper (Ref. 23), we found that adding small amount (~20%) of excess MAI can help compensate the loss of MAI and suppress the formation of PbI₂ during thermal annealing for perovskite grain growth. The XRD results in Figure 2f do not show any major compositional difference. To further verify the final composition particular locally, as requested by Reviewer 2, we have conducted energy-dispersive x-ray spectroscopy (EDS) study of these two types of perovskite films. The EDS line

scan has been often used in literature as previously shown by Ginger et al (Ref. 40). The EDS line scan results are shown in the new Supplementary Information Figure S3, showing no clear correlation between the Pb and I distribution and the grain morphology (e.g., grain versus grain boundary). For both samples, Pb and I stay roughly constant across multiple grains within experimental noise. The EDS analysis of the entire sampling area shows that the ratio of I/Pb is about 3.0 ± 0.3 for the small grain sample and 2.9 ± 0.3 for the large grain sample. The uncertainty is caused by the EDS detection limit (about 0.5–1%). Thus, it is reasonable to conclude that these two samples have essentially the same composition within experimental error and no local variations are observed across multiple grains.

Changes made: In order to address this valuable suggestion, we have included two sentences on Page 6 of the revised manuscript “In addition, energy-dispersive x-ray spectroscopy (EDS) measurement (Figure S3) also suggests ... Pb and I stay roughly constant across multiple grains within experimental noise and the ratio of I/Pb is essentially the same for both samples.” Details of the EDS data are found in Figure S3 and its caption. Finally, we acknowledge that the conclusion only holds up to the noise floor of our characterization tools. Following the Reviewer’s instruction, we add “Within the experimental errors of our characterization tools” in Materials and Methods (Page 12) for an appropriate statement.

Referee comment #2-3: I am not entirely satisfied with the response to my original point #2-3 on the impact of traps. I agree that techniques probing mobile carriers and photoluminescence in principle can be probing different population sub-sets. However, what is true is that mobile carriers and radiative recombination will both be impacted by the presence of unfilled trap states (in different ways but still generally negatively impacted). It is generally agreed that these trap states are unfilled at around 1 sun at CW steady-state illumination and even up to 10-100 sun (where they become filled and effects saturate), so over the range of excitation densities used here traps will play a large role. Recent time-resolved microwave conductivity measurements give further evidence to the impact of traps on conductivity (eg. doi: 10.1021/acs.jpcclett.5b01361). Is it simply a coincidence that the excitation-dependence microwave response in this work follows a very similar trend to the excitation-dependent photoluminescence trends? I think there must be a connection between them (i.e. the impact of traps) and this should at least be discussed and considered in the authors’ interpretations.

Reply #2-3: We thank the Reviewer for bringing up the time-resolved microwave conductivity (TRMC) result, which is now included as Ref. 46 in the revised paper, to our attention. Indeed, mobile carriers and radiative recombination are generally negatively impacted by the presence of unfilled trap states. As pointed out by the Reviewer, the excitation-dependence conductivity in Figure 2c follows a very similar trend to the filling of trap states. Therefore, we agree with him/her that traps states may play an important role in the measured photoconductivity data. Accordingly, we add two sentences on Pages 8 – 9 of the revised manuscript to discuss this possibility “On the other hand, it is generally believed that ... It is thus possible that trap states also play a role in the saturation of photoconductivity in Figure 2c.”

Reply to Referee 3’s report:

We respectfully disagree with the overall Reviewer's comments on our work. In our reply and revision in the last round, we did not "argue on every comments ... and did not actually address them". Instead, we made every effort to address the criticisms. Second, we totally acknowledge the fact that "this topic covers a very debated question". In the rapidly evolving perovskite solar cell research field, many topics are under lively debates. We believe that it is common practice to have such debate through literature. In our opinion, to make a statement like "I do not see a strong evidence from this study ...", the Reviewer should scrutinize the manuscript and raise **specific technical questions that are pertinent to our measurement**, as Reviewers 1 and 2 have done. Otherwise, the criticisms raised by the reviewer cannot be answered properly by our measurements. Third, the statement that our work "relied on a single measurement without considering much about material science" is not justified. On the contrary, besides the light-stimulated MIM, we have performed many experiments (IPCE, PCE, TRPL, XRD, EDS) to address the material science aspect of our films.

To summarize, by introducing a novel technique, carefully analyzing the experimental data, and providing the necessary supporting information based on a wide variety of material characterization techniques, we believe our result represents "an excellent piece of work" (Referee 1) rather than "a very preliminary study" (Referee 3).

In this round of review, the Reviewer raised three generic questions regarding surface effect, crystallinity, and defect states. Undoubtedly, these are important topics for the entire perovskite research field. However, the comments are still phrased without concerning our experimental details. In the following, we provide answers to individual questions from Reviewer 3.

Referee comment #3-1: The authors never consider the effect of film surface. It is well established in this field that the surface of perovskite films are very defective. It is true that a high efficiency over 18% can be made with these defective surfaces, because most of charge transport layers have passivation effect. The film used for this study is only 100 nm. The authors can have a simple estimation how fast the carriers (or maybe hot carriers) can diffuse to the surface and then be quenched based on the mobility they measured. There is no much difference of carrier mobility in on single grain and single crystals. If the surface recombination cannot be excluded, the authors cannot see any difference in the grains and grain boundaries, because surface recombination can be faster than that at the grain boundaries.

Reply #3-1: We are aware of the general concern about the surface recombination in these perovskite thin films. In a recent time-resolved microwave conductivity (TRMC) study coauthored by one of us (K. Zhu) using similarly prepared perovskite thin films with different grain sizes (Ref. 47), we found that the TRMC mobility shows a clear dependence on the grain size following a simple Kubo relation (Figure 2 in Ref. 47). This suggests that the carrier mobility of our perovskite thin films is not dominated by the surface properties of these films. More importantly, while the surface recombination may exist in these samples, it is *not* the dominating effect in our MIM measurement. In Figure 1b and the related discussion in our manuscript, we have clearly stated that "the span of the quasi-static electric field, which determines the lateral resolution and vertical probing depth, is set by the tip diameter" on Page 7. In other words, the MIM is a semi-surface tool that integrates the sample response from a volume of $\sim (100 \text{ nm})^3$ rather than from the surface layer. Even if the carriers quickly diffuse to the surface and become quenched there, the microwave electric fields can easily penetrate

through the surface and probe the entire depth of the film. We believe that it is distracting to the readers if we elaborate on this effect in our paper.

We emphasize that a certain experimental technique can only probe some physical properties of a sample, not all possible effects that may occur in the materials. One can name many other properties of the perovskite films (e.g. surface recombination) that simply do not affect our measurements in an appreciable manner. While the surface property mentioned by the Reviewer is relevant for perovskite film performance in general, it is not interrogated by our measurements. The Reviewer seems to be very concerned about some properties of the perovskite films likely influenced by his/her experience on these materials. However, these properties are not what our experiments are designed to investigate.

Referee comment #3-2: If the author wants to make a claim on discovery of correlation between crystallinity and electrical properties, the authors should have strong evidence on the crystallinity change of grains. Now every claim is based on assumptions. Regular XRD generally gives grain size information, but cannot tell defect density in bulk grains.

Reply #3-2: We believe that there is some misunderstanding. We are not trying to make a claim on discovery of correlation between crystallinity and electrical properties. This is indeed a challenge for the entire perovskite research field because there is no good way in literature to precisely quantify the degree of crystallinity of perovskite thin films. Thus, we can only qualitatively describe the crystallinity of perovskite thin films based on their XRD intensity and grain size. It is worth noting that we used a similar film thickness and thus a comparable amount of perovskite materials for such comparison. The XRD measurement condition is also fixed in the comparison. This qualitative comparison has been frequently used in the research field. The key result in our work is that, given the different grain sizes and PCE/TRPL/XRD data of the two films, we performed light-stimulated MIM experiments and obtained different photoconductivity maps. We believe that we have honestly reported the experimental findings and our claims are solid.

Referee comment #3-3: I do not see how the authors can reconcile the conflicting of the statement that defects in bulk affect material property but not at grain boundary. From fundamental material science, the defects in grain boundary is a collection of point defect in the bulk.

Reply #3-3: We feel that Reviewer 3 misunderstood our statement regarding the defects in the bulk versus defects at the grain boundary. We agree with him/her that the defects at grain boundary could be a collection of point defects in the bulk. However, we are not attempting to identify the chemical/physical natures of these defects in the current study; this has been a challenge for the entire perovskite research field. We acknowledge that there are three primary spatial locations of defects related to perovskite thin films, i.e., film surface, bulk of the grain, and boundary between neighboring grains and our technique is not able to differentiate the spatial location of these defects. Instead, our technique measures spatial variations of photoconductivity

Changes made:

On Page 3 of the manuscript, this point has been acknowledged as “various new growth controls ... could also affect ... defect density at the surface and in the bulk, and reduced structural defects ...”. In any case, the MIM measures the spatial variation of the photoconductivity, which does not differentiate the nature of defects.

With all their comments addressed, we hope that the manuscript is now ready for publication in *Nature Communications*. Thank you very much for your consideration.

Best regards,

Kai Zhu, Xiaoqin Li, and Keji Lai

Reviewers' comments:

Reviewer #1 (Remarks to the Author):

Chu et al. have presented a highly novel paper using an emerging scanning probe technique that allows to measure microwave absorption with sub-micron resolution. It is carefully written. The data is very thoroughly analysed and presented. I highly recommend the publication of this paper in Nature Communications.

Reviewer #2 (Remarks to the Author):

I am satisfied with the changes and can recommend publication.

Reviewer #3 (Remarks to the Author):

First of all, I want to reiterate that the authors misinterpreted the review comments. The reviewer did raise very specific questions. It is true that the authors used IPCE, PCE, TRPL, XRD, EDS to address the material science aspect of our films, however none of these routine technique can contribute to the main conclusion of this manuscript on the "benign grain boundaries". It is common to have different opinions in literature, but each work should be solid. The reviewer respectfully donot agree these questions are general or not related. These questions actually question the methodology the author used in this manuscript and the logic of manuscript in making the inappropriate conclusion. Let me explain why I think the conclusion made in this manuscript is very flawed again:

1. This manuscript has this major claim in the abstract: "The intrinsic photo-response is largely uniform across grains and grain boundaries, which is direct evidence on the benign nature of microstructures in these perovskite thin films." The authors actually measured the photoconductivity at grain and grain boundary, and come to the conclusion that grain boundaries are benign because of the uniform photoconductivity observed. However, the authors refused to consider my suggestion of considering the defective film surface. The equation 1 in this manuscript described the photoconductivity is determined by the product of carrier mobility and carrier recombination lifetime. The authors established the carrier mobility is not changed. If the perovskite film surface is very defective, as been reported by many other literatures, the uniform photoconductivity observed here is the result of a same carrier recombination lifetime at grain and grain boundary region, because photogenerated carriers can be quickly quenched by the defects at film surface, because of the very thin film used (100 nm). As long as the surface defects have stronger quenching capability than the grain boundary defect, the authors donot expect to see difference of conductivity at grain boundary and grain area. That is saying, the authors still cannot make a conclusion on the grain boundaries are benign, since the technique introduced in this method cannot distinguish recombination at surface or grain boundaries.

2. The authors gave conflicting statements on the major claims: On one hand, the authors responded: "We are not trying to make a claim on discovery of correlation between crystallinity and electrical properties." On the other hand, the abstract has this major conclusion. "In contrast, the carrier mobility and lifetime are strongly affected by bulk properties such as the sample crystallinity."

3. While the authors cannot make difference of bulk defects and grain boundary defects, the conclusion made is thus fundamentally wrong to any material scientist.

4. There are more publications recently showing that the perovskite grains may contain multiple domain boundaries, such as twin boundaries. Regular SEM of AFM cannot see these boundaries. So the last piece of claim in the abstract is not solid either: "As visualized by the spatial evolution of local photoconductivity, the degradation due to water diffusion through the capping layer begins with the

disintegration of large grains rather than the nucleation and propagation from grain boundaries.". Overall, almost every major claim in this piece of work is very flawed. I really hope the authors choose to publish the review comments so that that the readers understand the concern of the readers.

Dear Editor,

Thank you very much for sending us the review reports of our manuscript NCOMMS-17-08434A “Impact of Grain Boundaries on Efficiency and Stability of Organic-Inorganic Trihalide Perovskites”. We are very excited that both Referees 1 and 2 can now recommend the publication of our work after the revision. We are also glad that Referee 3 has provided specific comments for us to address. Detailed responses are listed below.

Reply to Referees 1 and 2’s reports:

We appreciate that both Reviewers have recommended our paper to be published in *Nature Communications* as is. Our manuscript has improved greatly after addressing their questions and comments.

Reply to Referee 3’s report:

We thank the Reviewer for raising specific questions in his/her comments. We now understand his/her concerns better and provide additional measurements to examine the role of the surface. In addition, we carefully rephrased our major findings to be more explicit about the properties that our novel scanning probe technique is capable of interrogating.

Referee comment #3-1: This manuscript has this major claim in the abstract: “The intrinsic photo-response is largely uniform across grains and grain boundaries, which is direct evidence on the benign nature of microstructures in these perovskite thin films.” The authors actually measured the photoconductivity at grain and grain boundary, and come to the conclusion that grain boundaries are benign because of the uniform photoconductivity observed. However, the authors refused to consider my suggestion of considering the defective film surface. The equation 1 in this manuscript described the photoconductivity is determined by the product of carrier mobility and carrier recombination lifetime. The authors established the carrier mobility is not changed. If the perovskite film surface is very defective, as been reported by many other literatures, the uniform photoconductivity observed here is the result of a same carrier recombination lifetime at grain and grain boundary region, because photogenerated carriers can be quickly quenched by the defects at film surface, because of the very thin film used (100 nm). As long as the surface defects have stronger quenching capability than the grain boundary defect, the authors do not expect to see difference of conductivity at grain boundary and grain area. That is saying, the authors still cannot make a conclusion on the grain boundaries are benign, since the technique introduced in this method cannot distinguish recombination at surface or grain boundaries.

Reply #3-1:

We thank the Reviewer for clarifying his/her suggestion of considering the defective film surface. As a control experiment, we have conducted the MIM experiments on a thicker (~ 300 nm) perovskite film made from the same 18% PCE material. The results are discussed in the resubmitted main text and detailed in Figure S9 of the SI. The MIM data are very similar to that on the thinner (~ 100 nm) sample shown in Figure S7. In particular, the MIM-Re images, which are less affected by the topographic crosstalk than the MIM-Im images, are again uniform across grains and grain boundaries. We would like to emphasize that the film thickness is comparable to that of the actual solar cell devices (~ 350 nm) demonstrating a PCE of $\sim 18\%$ in Figure S1. The new results provide direct evidence that the possible defective surface of the 100 nm film is not the cause of the observed uniform photoconductivity.

In addition, we would like to provide another piece of evidence that the surface recombination process is insignificant in our films. The following figure (unpublished) shows the TRPL results using two-photon (1090 nm) excitation and one-photon (405 nm and 640 nm) excitation on a thick (~ 1 μm) perovskite film. These different wavelengths will lead to different carrier generation profiles across the film thickness due to the different light penetration depths. Based on the absorption spectrum, we estimated that the absorption coefficient α is about 43 nm^{-1} and 172 nm^{-1} at 405 nm and 640 nm, respectively. Thus, the optical penetration length ($1/\alpha$) is about 43 nm and 172 nm when illuminated at 405 nm and 640 nm, respectively. In contrast, the two-photon experiment probes the entire 1- μm film because of much weaker absorption coefficient. In general, the TRPL lifetime (τ_{TRPL}) can be affected by the bulk lifetime (τ_{B}) and the surface recombination velocity S as given by the following expression: $\tau_{\text{TRPL}}^{-1} = \tau_{\text{B}}^{-1} + \alpha S$. If the surface recombination dominates the recombination process, then $\tau_{\text{TRPL}} \approx 1/\alpha S$. Assuming S is independent of the incident wavelength, the lifetime should scale with the optical penetration length, which is expected to increase by a factor of 4 (or >20) when the excitation wavelength is increased from 405 nm to 640 nm (or 1090 nm). However, the following TRPL figure shows similar decay kinetics regardless of the excitation wavelength, indicating that surface recombination is not significant in the perovskite thin films used in this study.

[Figure was redacted here]

With that said, we feel that the inclusion of experimental details of TRMC (Ref. 47) and two-photon TRPL measurements will distract the readers from the focus of our paper. We are currently considering a separate paper on the detailed analysis of two-photon TRPL study of perovskites. If possible, we would prefer to keep this interesting discussion, especially the

unpublished data in the response letter. We will be glad to publish the rest of the communication as suggested by the Reviewer.

Changes made:

1. A new section describing the MIM results on a 300 nm MAPbI₃ film (18% PCE sample) is now included in Figure S9. The images are indeed very similar to those acquired on the 100 nm films, indicating that the possible surface recombination process is not the cause of the observed uniform photoconductivity. Accordingly, a sentence is added to Page 10 of the main text as “As a control experiment, we have also conducted the MIM measurement on a thicker ($H = 300$ nm) film of the 18% PCE sample and observed the same results (Figure S9), which indicates that the MIM signals are not dominated by the possible surface recombination in the 100 nm film.”
2. We believe that the phrase “the grain boundaries are benign” is understood differently by the Reviewer from our intention. We agree that it is more appropriate to make a statement specific to our experimental results in the Abstract. In the revised manuscript, the sentence pointed out by the Reviewer in the Abstract has been changed to “The microwave signals are largely uniform across grains and grain boundaries, which indicates that microstructures in these perovskite thin films do not lead to strong spatial variations of the intrinsic photo-response”. By removing the claim of “direct evidence on the benign nature”, we believe that the abstract is now a precise description of the experimental data.

Referee comment #3-2: The authors gave conflicting statements on the major claims: On one hand, the authors responded: “We are not trying to make a claim on discovery of correlation between crystallinity and electrical properties.” On the other hand, the abstract has this major conclusion. “In contrast, the carrier mobility and lifetime are strongly affected by bulk properties such as the sample crystallinity.”

Reply #3-2: We believe that this is simply a misunderstanding. For the sentence quoted by the Reviewer from our previous response letter, “correlation” refers to a definitive connection between crystallinity and electrical properties, which is not possible based on our measurement. Somehow, the sentence was interpreted out of context. The next sentence in our reply letter reads as “This is indeed a challenge for the entire perovskite research field because there is no good way in literature to *precisely quantify* the degree of crystallinity of perovskite thin films”. With this context, we do not intend to make any claims that are beyond our experimental findings.

The quoted sentence in the Abstract is a description of our experimental finding on photoconductivity, what photoconductivity depends on, and how photoconductivity is influenced by the crystallinity of the samples we have investigated. In Figs. 2c and 2e, we show that the measured photoconductivity and lifetime of the 18% PCE sample are higher than that of the 15% PCE sample. In Fig. 2f, the XRD signals suggest that the crystallinity of the 18% PCE sample is better than the 15% counterpart. In terms of the difference between the two samples, we provided a thorough analysis to Reviewer 2’s comment #2-2 and established that the two films have essentially the same composition within experimental error (see Reply #2-2 in the previous response letter). We note that Reviewer 2 is now fully satisfied with the changes.

Changes made: In order to avoid further confusion, it is appropriate to strictly stick to the experimental data in our presentation. In the revised manuscript, we change “In contrast, the carrier mobility and lifetime...” to “In contrast, the measured photoconductivity and lifetime...” in the Abstract. With this modification, we hope that Reviewer 3 finds this revised statement robust.

Referee comment #3-3: While the authors cannot make difference of bulk defects and grain boundary defects, the conclusion made is thus fundamentally wrong to any material scientist.

Reply #3-3: First of all, we acknowledge that the MIM measurement does not make difference of bulk and grain boundary defects. If there is any literature reporting methods that are capable of differentiating surface and grain boundary defects in these materials, we would be glad to include the citation and discuss them in our paper. One of us (Zhu) tried electroluminescence measurements and found that the sample was quickly damaged. Many electrical and optical experiments including ours are designed to detect the material response to external excitations. These experiments cannot answer this particular question that the Reviewer found critical.

Changes made: We fully understand the Reviewer’s concern about the different types of defects in the perovskite films. As stated in Reply #3-1, we have removed “direct evidence on the benign nature” from the Abstract and used the more appropriate wording of “The microwave signals are largely uniform across grains and grain boundaries...”.

Referee comment #3-4: There are more publications recently showing that the perovskite grains may contain multiple domain boundaries, such as twin boundaries. Regular SEM or AFM cannot see these boundaries. So the last piece of claim in the abstract is not solid either: “As visualized by the spatial evolution of local photoconductivity, the degradation due to water diffusion through the capping layer begins with the disintegration of large grains rather than the nucleation and propagation from grain boundaries.”.

Reply #3-4: We thank the Reviewer for bringing up the recent progress on the imaging of twin boundaries. Indeed, since these boundaries are not visible in our AFM measurement, we cannot exclude the possibility that they are the nucleation centers of the degradation sites.

Changes made: We acknowledge the limitation of our experimental methods in the following ways. (1) In that particular sentence in the Abstract, we limit our discussion to “... the nucleation and propagation from grain boundaries observable in the topographic image.” (2) On page 11 of the main text, we include two sentences to discuss the twin boundaries – “We note that recent transmission electron microscopy (TEM) studies have revealed the existence of twin boundaries that are not seen in traditional SEM or AFM images⁵². Our results are thus only valid for the GBs with clear topographic features.” (3) A new reference to *Nat. Commun.* **2017**, *8*, 14547 by Yi-Bing Cheng’s group is added to the bibliography. The paper is now solid after these revisions.

Referee comment #3-5: Overall, almost every major claim in this piece of work is very flawed. I really hope the authors choose to publish the review comments so that that the readers understand the concern of the readers.

Reply #3-5: We agree that publishing the review comments along with this paper so that the readers can appreciate the revisions throughout the review process. Our only concern is on the

unpublished data, which is a critical piece of information for a new paper that one of us (Zhu) and other collaborators are working on.

To summarize, we are thrilled to see the fully supportive review reports from both referees 1 and 2. We have also made our best effort to address the comments from referee 3 and modified the manuscript accordingly. As pointed out multiple times by referees 1 and 2, our work “using an emerging scanning probe technique” is “highly novel” and the data are “very thoroughly analyzed and presented”. We sincerely thank you for your consideration and support the acceptance of this work in *Nature Communications* in a timely manner.

Best regards,

Kai Zhu, Xiaoqin Li, and Keji Lai

REVIEWERS' COMMENTS:

Reviewer #3 (Remarks to the Author):

I appreciate the authors' extra efforts to address the comments. I can support the publication of this manuscript, but hope the authors appropriately acknowledge these facts in their manuscript, and best publish the comments after removing their unpublished results.

Surface recombination: the two-photon TRPL data provided clearly showed influence of surface recombination. The authors ignored that both surfaces (top and bottom) can quench charges, and thus one would not see huge difference in PL lifetime for a film with 1 micrometer thick. However even with this thickness, the data provided still show difference in charge recombination lifetime. The device efficiency of 18.3% cannot exclude the presence of a large surface defect density, because there are always charge transport layers in real devices which passivate the defects. If the devices are made with direct metal contact, the devices generally are very bad. Actually the method for the films made in this manuscript is pretty general in this field. It is hard to believe the films made by the same method have low surface defect density. I would suggest the authors not exclude the possible effect of surface recombination, because the evidence the authors provided cannot.

Degradation: I would suggest the authors to remove the statement in the abstract, because such a superficial statement should not be published anywhere.

Dear Editor,

Thank you very much for sending us the review reports of our manuscript NCOMMS-17-08434B “Impact of Grain Boundaries on Efficiency and Stability of Organic-Inorganic Trihalide Perovskites”. We are very excited that Referees 3 can now recommend the publication of our work after minor revisions. Our responses to specific comments from Referee 3 are list below.

Reply to Referee 3’s report:

We appreciate the reviewer’s support for publication of our manuscript. This paper has been greatly improved after considering and addressing his/her concerns and comments in the past rounds. We also agree with the reviewer to publish the review reports and our rebuttal letters for the best interest of the readers.

Referee comment #3-1:

Surface recombination: the two-photon TRPL data provided clearly showed influence of surface recombination. The authors ignored that both surfaces (top and bottom) can quench charges, and thus one would not see huge difference in PL lifetime for a film with 1 micrometer thick. However even with this thickness, the data provided still show difference in charge recombination lifetime. The device efficiency of 18.3% cannot exclude the presence of a large surface defect density, because there are always charge transport layers in real devices which passivate the defects. If the devices are made with direct metal contact, the devices generally are very bad. Actually the method for the films made in this manuscript is pretty general in this field. It is hard to believe the films made by the same method have low surface defect density. I would suggest the authors not exclude the possible effect of surface recombination, because the evidence the authors provided cannot.

Reply #3-1:

We thank the reviewer for clarifying the effect of surface recombination. We agree that the surface effects cannot be excluded in our MAPbI₃ films and have made the changes as follows.

Changes made:

We acknowledge the possible effect of surface recombination in the following ways. 1) We deleted the sentence “The reduction of bulk and surface defect densities and the improvement of sample crystallinity are clearly important for future material and device optimization.” from the main text (Line 178) to avoid possible confusion and misunderstandings. 2) We added the following statement in Supplementary note #1 to acknowledge the possible effect of surface defects as well as their role in intrinsic photo-response – “The results suggest that the MIM signals on both the thin (100 nm) and thick (300 nm) films in our study are not dominated by the surface recombination effect. However, we also acknowledge that surface defects are generally present in MAPbI₃ devices and their effects cannot be totally excluded in our measurements.”

Referee comment #3-2:

Degradation: I would suggest the authors to remove the statement in the abstract, because such a superficial statement should not be published anywhere.

Reply #3-2:

Our statement in the Abstract should honestly reflect the measurement results. As a result, we prefer to follow the Editor's suggestion, i.e., to keep the main claim on the degradation mechanism and clarify what we mean by 'grain boundaries'.

Changes made:

We revised the statement in the abstract as follows – "... the degradation process begins with the disintegration of grains rather than nucleation and propagation from visible boundaries between grains." To keep the abstract within 150 words, we believe this is the best way to distinguish the large visible grain boundaries from the boundaries inside the grains. On Page 11 of the main text, we fully describe the difference between the two types of boundaries – "We note that recent transmission electron microscopy (TEM) studies have revealed the existence of twin boundaries that are not seen in traditional SEM or AFM images⁵¹. Our results are thus only valid for the GBs with clear topographic features." We believe this arrangement is sufficient for the readers to appreciate our summary statement in the abstract.

To summarize, we sincerely thank all reviewers and the editor for their efforts, comments, and suggestions, which have greatly improved the quality of this manuscript towards its publication in *Nature Communications*.

Best regards,

Kai Zhu, Xiaoqin Li, and Keji Lai